# Remote Sensing Analysis of Typhoon-Induced Storm Surges and Sea Surface Cooling in Chinese Coastal Waters

Xiaohui Li [1], Guoqi Han [2], Jingsong Yang [1,3,*] and Caixia Wang [4]

1   State Key Laboratory of Satellite Ocean Environment Dynamics, Second Institute of Oceanography, Ministry of Natural Resources, Hangzhou 310012, China
2   Fisheries and Oceans Canada, Institute of Ocean Sciences, Sidney, BC V8L 4B2, Canada
3   Southern Marine Science and Engineering Guangdong Laboratory (Zhuhai), Zhuhai 519082, China
4   Physical Oceanography Laboratory, Ocean University of China, Qingdao 266100, China
*   Correspondence: jsyang@sio.org.cn

**Abstract:** Inthis study, remote sensing measurements were utilized to examine the characteristics of storm surges and sea surface cooling in Chinese coastal waters caused by typhoons. Altimetric data from satellite altimeters were used to determine the magnitude, cross-shelf decaying scale, and propagating speed of storm surges from typhoons. The results were in agreement with estimates obtained from a theoretical model and tide gauge data, showing that the two storm surges propagated as continental shelf waves along the southeastern coast of China. The sea surface cooling, driven by Typhoons 1319Usagi and 1323Fitow, was analyzed using the remote sensing sea surface temperature product, named the global 1 km sea surface temperature (G1SST) dataset, revealing a considerable decrease in the temperature, with the largest decrease reaching 4.5 °C after the passage of 1319Usagi, in line with buoy estimates of 4.6 °C. It was found that 1323Fitow and 1324Danas jointly impacted the southeastern coast of China, resulting in a significant temperature drop of 4.0 °C. Our study shows that incorporating remotely sensed measurements into the study of oceanic responses to typhoons has significant benefits and complements the traditional tide gauge network and buoy data.

**Keywords:** satellite altimetry; typhoons; storm surge; sea surface cooling





## 1. Introduction

Intense synoptic events of the climate system, known as tropical cyclones (TCs), may disrupt currents and result in storm surges that inflict immediate damage to coastal communities [1–3]. Storm surges caused by TCs are some of the worst natural disasters and are the primary causes of exceptionally high sea levels and catastrophic floods [4–6]. In addition, TCs introduce a substantial amount of mechanical energy into the water, which has a range of impacts on the ocean. Previous studies confirmed the mixing of cold- and nutrient-rich water into the surface layer, the air–sea interaction of moisture and heat from the sea surface, and the subsurface change with a cold or warm anomaly [2,3,7–9]. After the passage of typhoons or hurricanes, phytoplankton blooms were also induced along the TC tracks [2,10–12]. Considering the intensification of tropical cyclones due to global warming, conducting remote sensing analyses of tropical cyclone-induced storm surges and sea surface cooling is becoming increasingly important.

Traditionally, observing and comprehending the characteristics of storm surges has been conducted through the analysis of tidal gauge data, as it is considered the most reliable method according to previous research [5]. However, due to the limited number of tide gauges, it is challenging to precisely determine the detailed spatial distribution of storm surges. To overcome this limitation, numerical models have been utilized to provide real-time spatiotemporal characteristics of storm surges, allowing for a better understanding of the mechanisms involved [13,14]. The use of satellite remote sensing to observe the ocean has been a significant advancement in oceanography [15]. With the vast improvement

in temporal and spatial coverage of the ocean provided by satellite observations, ocean phenomena can be observed and understood more comprehensively.

Historically, the use of satellite observations to investigate the physical mechanisms behind the oceanic responses to typhoons or hurricanes has been significant. Hajji and Olagnon [16] extracted the storm surge residual from satellite measurements in the North Sea, and Scharroo et al. [17] reported using satellite altimetry to detect storm surges. An increasing number of studies are recognizing the importance of satellite observations, as demonstrated by our previous study [18], which showed that Jason-2 and tide gauge data simultaneously observed the storm surge caused by Hurricane Igor, and satellite altimetry measurements revealed a correlation between a storm surge and continental shelf waves. Additionally, Lillibridge et al. [19] and our previous study [20] analyzed the Hurricane Sandy storm surge observed by the HaiYang-2A (HY-2A) satellite altimeter, and our previous study [21] showed that altimetry played a crucial role in improving the accuracy of storm surge predictions. Ji et al. [5] conducted a statistical analysis of data from tide gauges and satellite altimeters used to monitor storm surges in China's coastal regions. While the success of altimeter satellites in observing typhoon-induced storm surges is highly stochastic, the use of altimeter data is still valuable for providing timely and accurate information for monitoring these events. Storm surges are short-lived and highly destructive events that can cause significant damage to human societies. Given the severe impacts of storm surges, high temporal resolution data are essential for effective monitoring and mitigation. The use of altimeter data is valuable for timely and accurate monitoring of storm surges along the coast of China, where typhoons frequently occur.

To assess the impact of passing typhoons and hurricanes on the upper ocean, various methods have been employed, such as sea surface temperature (SST) measurements from buoys, moorings, remote sensing satellites, and numerical models [10,19,22–31]. For instance, the effects of Typhoon Megi were studied using data from the Argo buoy and reanalysis [32]. The relationship between SST and the sea surface wind speed during typhoons over the South China Sea was also investigated [33]. According to remote sensing observations, the highest TC-induced surface cooling was around 7 °C [34]. Multiple data sources were used to investigate the changes in the upper ocean and subsurface during the Roanu cyclone in May 2016 [35]. In the northern hemisphere, the strongest cooling regions were often on the right side of TC tracks [28,32,35]. It was recognized that the magnitude of TC-induced surface cooling was influenced by various factors, such as the intensity and translation speed of TCs, and the oceanic conditions prior to their passage, such as mesoscale eddies [24,36,37]. The presence of mesoscale eddies can have a significant impact on the dynamics of the ocean-atmosphere system, which can impact the magnitude and spatial patterns of SST variations along the cyclone track and the strength of tropical cyclones [38–40]. Despite the limitations of microwave radiometers in accurately retrieving SSTs during typhoon periods due to the impact of rain and clouds, the fusion of SST products can provide a better resolution and may be more effective in analyzing the SST response to typhoons or hurricanes. However, it should be noted that the fusion process itself can introduce errors. Therefore, when using fusion SST, it is necessary to consider the advantages and disadvantages between it and a single remote sensing observation SST, as well as select the appropriate data source according to the specific application scenario. With the improvement of remote sensing technology and data fusion techniques, the accuracy of satellite-based SST products in coastal areas is gradually improving with a high resolution of up to 1 km. This makes it possible to analyze the SST response to typhoons or hurricanes.

The objective of this study is to investigate the dynamic and thermal responses of the upper ocean to two typhoons: Typhoons 1319Usagi and 1323Fitow. The Jason-2 altimeter satellite and multi-satellite SST remotely sensed data were utilized to examine the impact of typhoons on storm surges and sea surface cooling, as shown in Figure 1, which depicts the Jason-2 ground track, tide gauges, buoy, and the typhoon tracks. The Jason-2 satellite, using track-88 and track-240, observed the storm surge induced by 1319Usagi and 1323Fitow,

respectively. The Jason-2 altimeter data were used to analyze the magnitude of the storm surge, its cross-shelf e-folding scale, and phase speed. These findings were also validated through observations from tide gauges and theoretical models of shelf waves. In addition, global 1 km SST (G1SST) data and buoy observations were used to investigate the cooling of the sea surface after the typhoons passed along the southeast coast of the China Sea.

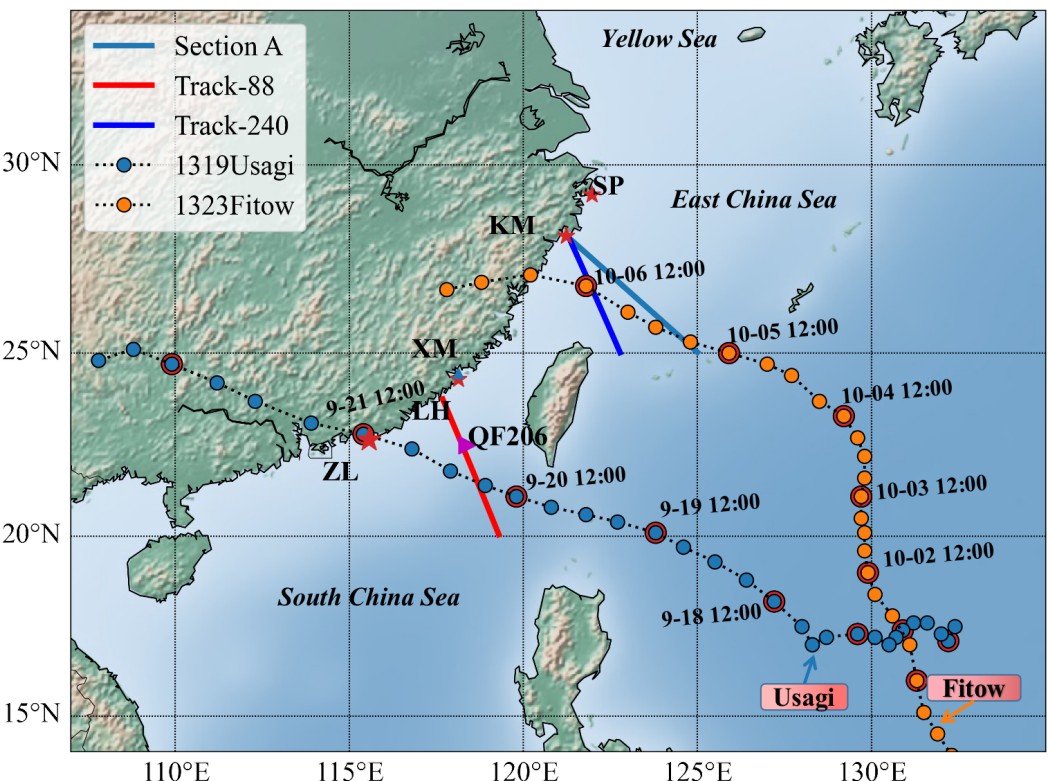

**Figure 1.** Geographical distribution of satellite ground tracks of Jason-2 track-88 (red line) and track-240 (blue line), coastal tide gauges (star), QF206 buoy location (magenta triangle), and the tracks of Typhoons 1319Usagi and 1323Fitow. ZL: Zhelang, LH: Longhai, XM: Xiamen, KM: Kanmen, SP: Shipu.

## 2. Data and Method

### 2.1. Satellite Altimetry Data

In this study, the sea surface height anomaly (SSHA) data from Jason-2 were obtained from the Archiving Validation and Interpolation of the Satellite Oceanographic Data Center (AVISO) and the Center for Topographic Studies of the Ocean and Hydrosphere (CTOH, accessed on 31 May 2022). The along-track SSHA data have a spatial resolution of 6~7 km, and the satellite ground track was repeated every 10 days. The CTOH provides access to its frequently updated altimetric databases, including TOPEX/Poseidon, Jason-1, Jason-2, GFO, and Envisat data using X-TRACK software [41,42], which contains a relatively larger number of valid data points than those generated by the standard geophysical data record (GDR) [43]. The SSHA estimates are observed more than 10 km from the coastline in AVISO, while those derived from X-TRACK are closer to land, at around 5 km [43]. This provides more effective data for observing the storm surge in areas closer to the shore.

Data on SSHA from Jason-2 (CTOH L3 TPJ1J2 product) were obtained for the ground tracks of track-88 (red line) and track-240 (blue line) as shown in Figure 1. Equation (1) was applied to process a series of geophysical and environmental corrections, including instrument calibration (*inst*), ionospheric correction (*iono*), tropospheric correction (*trop*, dry and wet tropospheric correction), sea state bias correction (*bias*), tidal correction (*tide*; solid earth, pole, and loading tides), and dynamic atmospheric correction (*DAC*) in the



CTOH L3 product [43]. As the sea surface height anomaly caused by atmospheric dynamic effects should be taken into consideration during the typhoon, the *DAC* correction term was not subtracted.

$$SSHA = ALT - R - (inst + iono + trop + bias + tide + DAC) - MSS \qquad (1)$$

where *ALT* is the orbital height of the satellite, *R* is the distance between the instrument and the mean reflected sea level, and *MSS* is the mean sea surface height.

The longwave Equations (2) and (3) based on LeBlond and Mysak's longwave theory were utilized to analyze the altimeter sea surface height anomalies [44]. Equations (2) and (3) demonstrate that the wave's amplitude decreases exponentially in the cross-shelf direction.

$$S(x) = S_0 + S_1 * \exp(-x/L) \qquad (2)$$

$$c = L * f \qquad (3)$$

where *S* is the sea surface height anomaly (unit: m), *x* is the distance in the cross-shelf direction (unit: m), if *x* = 0, then $S(0) = S_0 + S_1$, $S(0)$ is the SLA at the coastal location. *L* represents the cross-shelf decaying scale, *f* is the Coriolis parameter (unit: $s^{-1}$). Equation (3) can be used to calculate the wave-phase speed *c*, which has been successfully applied to extract the propagation of storm surges in [18,20].

Before fitting the altimeter data, it is necessary to apply a 50 km Butterworth low-pass filter (Equation (4) [19]) to suppress the range noise in the SSHA data obtained from the altimeter. Our previous studies indicated altimeters can extract the characteristics of storm surges based on Equations (2) and (3). In our current research, we found that Jason-2 continuously observed two typhoons in 2013, providing valuable observation data for our study. We attempt to extract storm surge characteristics to investigate the mechanisms of the two storm surges in Chinese coastal waters.

$$H_{lp}(f_0) = \frac{1}{1 + (\frac{f_0}{f_c})^{2n}} \qquad (4)$$

In this study, the cutoff frequency $f_c$ was set to 1/50 Hz, which means that wavelengths shorter than 50 km were attenuated by the filter. The filter order *n* is usually chosen based on the desired steepness of the cutoff, and in this study, a second-order filter was used. The altimeter data were sampled at a frequency of 1 Hz, as is typical for altimetry measurements. The sampling frequency $f_0$ was set to 1 Hz in the paper.

### 2.2. Theoretical Model of Coastally Trapped Waves

In order to estimate the propagation speed of coastally trapped waves, Chen and Su [45] established a relationship between coastal topography and the wave-phase speed by utilizing theoretical models of the coastal topography along the coast of the China Sea, including the Yellow Sea model, the East China Sea model, and the South China Sea model, as shown in Equation (5). The boundary conditions, along with the corresponding differential equation, form a Sturm–Liouville boundary value problem. Once the form of $H(x)$ is given, the eigenfunctions $F_n(x)$ and eigenvalues $c_n$ ($n = 1, 2, \ldots$) can be solved. This provides an effective method for estimating the propagation speed of coastally trapped waves along the coast of the China Seas. Moreover, it can also be used to validate the estimated wave velocity from the altimeter.

$$H \left( \frac{dF}{dx} + \frac{f}{c} F \right) = 0 \qquad (5)$$

where *x* is the distance in the cross-shelf direction, *F* is the distribution function of water level in *x* direction, *f* represents the Coriolis parameter, *c* is the propagation speed of coastally trapped waves.

In this study, the theoretical models of the East China Sea and South China Sea are used to calculate the propagation speeds of coastally trapped waves. We selected three sections to estimate the velocity of coastally trapped waves based on the location of the typhoon storm surge and the coastal topography, as shown in Figure 2. Expressions approximately describing the coastal topography, referred to as $H$, were obtained using ETOPO1 data for these two regions, as shown in Equation (6) for the East China Sea and Equation (7) for the South China Sea.

$$H(x) = \begin{cases} \alpha x, & 0 \le x < L_1 \\ h_1 + h_2, & L_1 < x \le L \end{cases} \tag{6}$$

$$H(x) = \begin{cases} \alpha x, & 0 \le x < L_1 \\ (h_2/L_2) * x + (h_1 - (h_2 * L_1)/L_2), & L_1 < x \le L_1 + L_2 \\ h_1 + h_2 & L_1 + L_2 < x \le L \end{cases} \tag{7}$$

where $\alpha = h_1/L_1$, $x$ is the distance in the cross-shelf direction, $L_1$ represents the length of the first section of the sea topography, and $L_2$ represents the length of the second section of the sea topography.

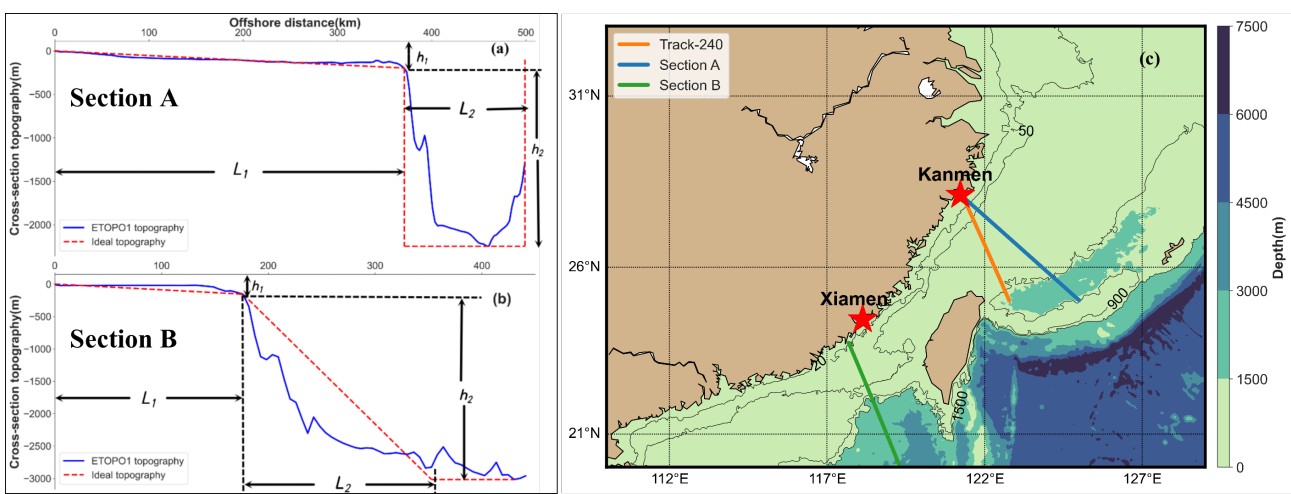

**Figure 2.** The cross-section topography for estimating the propagation speed of the coastally trapped wave. The ETOPO1 and simplified topography of (**a**) Section A in the East China Sea, (**b**) Section B in the South China Sea; (**c**) the blue line represents the position of the topographic section of the East China Sea; the green line represents the position of the topographic section of the South China Sea.

The reason for not selecting track-240 is due to the necessity of expanding water depth lines that are as perpendicular as possible to the water depth contour lines. Based on the analysis presented in Figure 2, the newly chosen cross-sectional profile aligns more closely with the requirements outlined by the theoretical model evaluation. Regarding in Figure 2a, the selection of $L_2$, which extends from the continental shelf to approximately 500 km offshore, was motivated by the focus of our study on the shelf waves and their propagation in the East China Sea, particularly in the region east of the Ryukyu Islands. This region boasts a unique topography, with the Ryukyu Islands serving as a barrier that separates the East China Sea from the Pacific Ocean, as described in [45]. Hence, in order to effectively capture the coastally trapped waves in this region, it was deemed necessary to extend the shelf region to 500 km offshore.

For the same considerations, we chose track-88 as the estimation of the propagation speed of the shelf waves in the southern South China Sea. In consideration of in Figure 2b, the determination of $L_2$ extending from beyond the deepest depth and not starting at the continental break was deliberate in light of the research focus on the shelf waves in the South China Sea. These theoretical models provide a basis for understanding the variation

in coastal topography and its effect on coastally trapped wave propagation in these two sea areas. The velocity estimated by the theoretical models can be used to validate the propagation speed estimated by the altimeter, enabling us to determine the propagation characteristics of the storm surge in the China Sea.

### 2.3. Tide Gauge and Buoy Data

During Typhoons 1319Usagi and 1323Fitow, hourly tide gauge observations were collected at eight locations, including Shipu (SP), Kanmen (KM), Xiamen (XM), Longhai (LH), and Zhelang (ZL) (as shown in Figure 1). The tidal heights involving 59 tidal constituents were predicted using a harmonic analysis [46]. The predicted tide height ($H_{tide}$) was then subtracted from the observed sea level data ($SL_{obs}$) to obtain the de-tided sea level anomaly ($SLA$), as expressed in Equation (8).

$$SLA = SL_{obs} - H_{tide} \tag{8}$$

The sea surface meteorological parameters, such as wind speed, wind direction, air temperature, pressure, sea surface temperature, and wave height, were measured using the buoy located at 118.4°E and 22.5°N (marked in magenta triangle in Figure 1). The use of this buoy allowed for continuous monitoring of important ocean and atmospheric conditions during the typhoons, providing valuable data for the investigation.

### 2.4. Satellite-Based SST Data

The G1SST data were obtained from the Group for High-resolution Sea Surface Temperature (GHRSST, https://www.ghrsst.org/, accessed on 31 October 2022). The G1SST data, created at 1 km ultra-high resolution, were integrated in real time with the SST data collected from various satellite sensors, such as the advanced microwave scanning radiometer for earth observing satellite (AMSR-E), moderate resolution imaging spectroradiometer (MODIS), advanced along-track scanning radiometer (AATSR), spinning enhanced visible and infrared imager (SEVIRI), multifunctional transport satellites (MTSAT), the advanced very high-resolution radiometer (AVHRR), and in situ observations. The high temporal and spatial resolutions of G1SST make it an ideal data source for investigating the cooling of the sea surface after the typhoons passed.

In order to assess the response of SST to typhoons, it is necessary to compute the daily sea surface temperature anomaly (SSTA), which is obtained by subtracting the daily climatological SST values from the corresponding daily SST values, as described in [35,47]. The daily climatological SST values were generated using a long-term dataset of SST measurements. In this study, the climatological daily SST values were computed as the average SST values for the years 2010–2013.

The pre-typhoon SSTA state is computed as the average SSTA over a specified period of time prior to the arrival of the typhoon, excluding the period of the cyclone. This establishes a baseline for the sea surface temperature against which the impact of the typhoon can be assessed. Mathematically, the pre-typhoon SSTA state is expressed as:

$$SSTA_{Pre} = \frac{1}{N_1} \sum_{i=1}^{N_1} SSTA_i \tag{9}$$

where $N_1$ is the number of days in the pre-typhoon period and $SSTA_i$ is the SSTA for day *i* in the pre-typhoon period, excluding any days within the period of the cyclone. Additionally, to remove the influence of the temperature on the day of the cyclone, the SST value on the day of the cyclone is also excluded from the calculation.

## 3. Results

### 3.1. Storm Surge Observed by Jason-2 and Tide Gauges

The Jason-2 altimeter satellite, the tide gauge stations in Xiamen, Longhai, and Zhelang, and the buoy (no. QF206) recorded the storm surge induced by 1319Usagi. Moreover,

1319Usagi made landfall near the Zhelang tide gauge station with a maximum wind speed of 45 m/s and central air pressure of 935 MPa. The de-tided sea level anomaly (SLA) data from Xiamen, Longhai, and Zhelang stations are displayed in Figure 3. The nonlinear interactions between the storm surge and astronomical tide refer to the complex interplay between the high water levels caused by the typhoon's winds and the normal tidal patterns. Additionally, the bathymetry and coastline configurations of the strait can also play a role in shaping the surge patterns, leading to different magnitudes and distributions of the storm surges along the coastlines as seen in the observations at the Xiamen, Longhai, and Zhelang tide gauge stations and depicted in Figure 3. This phenomenon was also noted in a previous study, such as the storm surge caused by Typhoon Seth [21]. The peak surge height at the Zhelang tide gauge station was approximately 1.63 m at 12:00 on 22 September, as shown in Figure 3c.

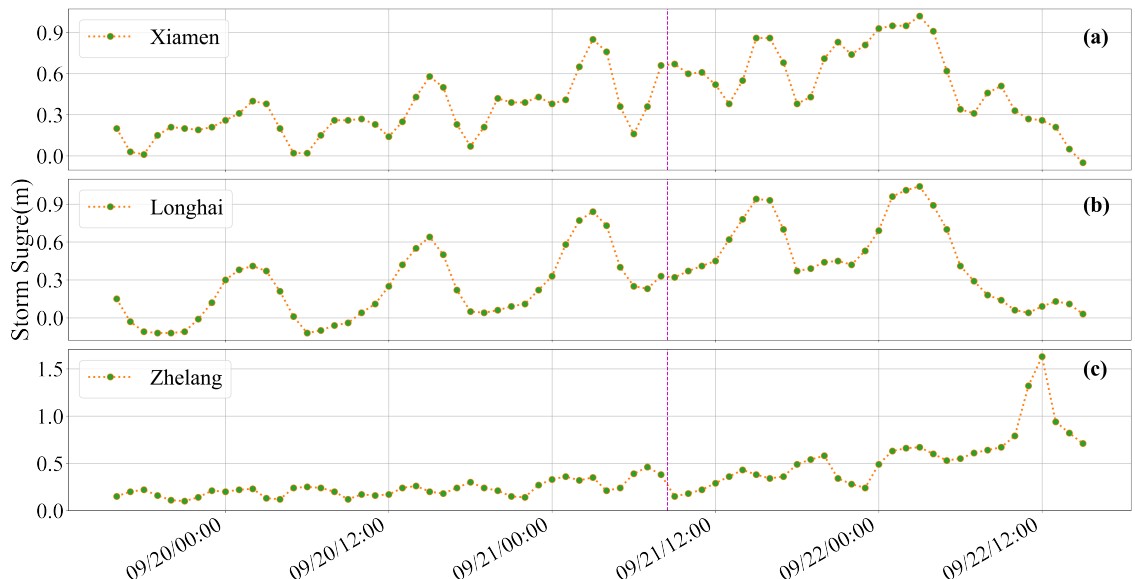

**Figure 3.** De-tided SLAs induced by 1319Usagi at (**a**) Xiamen, (**b**) Longhai, and (**c**) Zhelang tide gauge stations. The magenta dashed lines indicate the passing time of the Jason-2 satellite (08:29 UTC on 21 September 2013). The landing time of 1319Uasgi was 11:40 on 22 September.

Fortunately, the Jason-2 altimeter satellite passed over the South China Sea near Xiamen Fujian at 08:29 UTC on 21 September 2013, providing valuable information for the investigation of the storm surge. As the coastal data of cycle-773 were invalid in the CTOH L3 TPJ1J2 product, it was replaced by cycle-772. The SLAs of the Jason-2 altimeter track-88 from cycle-772, cycle-774, and cycle-775 are shown in Figure 4a. The variations in the magnitude of sea surface height anomalies before (cycle-772, 12:36 UTC on 1 September 2013) and after (cycle-775, 06:31 UTC on 1 October 2013) the storm were distinct, with a shift from the increasing trend of SLA recorded by Jason-2 during the storm (cycle-774). Additionally, the SLAs of cycle-774 significantly increased from the shelf break at 22°N toward the coast. Conversely, the SLAs of cycle-772 and cycle-775 were nearly zero near the coast and did not show a clear storm surge signal. The magenta dashed line in Figure 3a,b represents the passing time of the Jason-2 satellite. The closest measurement of SLA to the coastline (about 3 km away) by track-88 from cycle-774 of the Jason-2 altimeter was 0.55 m, while the measurements simultaneously made by the Xiamen, Longhai, and Zhelang tide gauge stations were 0.66, 0.33, and 0.38 m, respectively. We took the average of the observed values at the three tide stations, which was approximately 0.46 m; the observed value of the altimeter was in close agreement with this average.

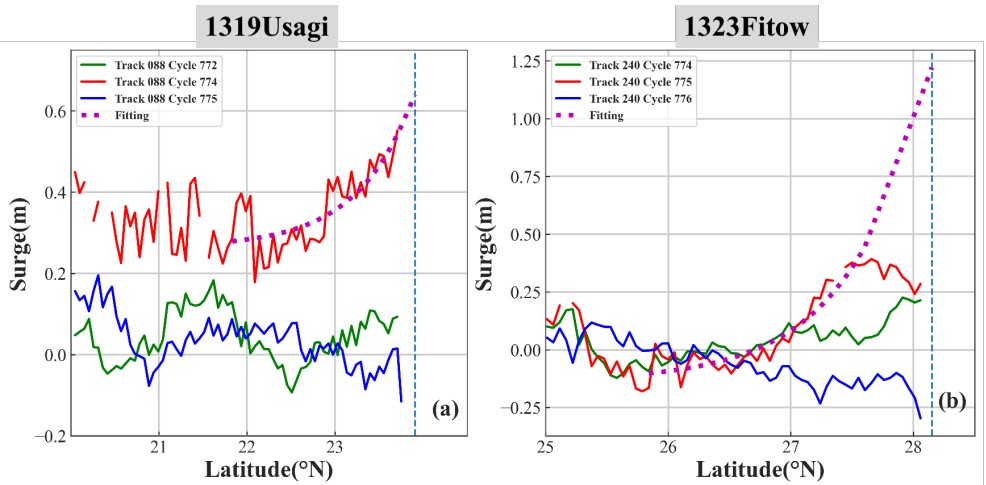

**Figure 4.** Sea surface height anomalies along the ground track were recorded by Jason-2 before (green line), during (red line), and after (blue line) the typhoon passed; fitted data are shown with magenta dashed lines. (**a**) Typhoon 1319Usagi; (**b**) Typhoon 1323Fitow. The coastline is represented with a dashed line. Data on SSHA from Jason-2 were obtained from the CTOH L3 TPJ1J2 product.

Similar to the analysis of 1319Usagi, the Jason-2 altimeter data and tide gauge observations were also used to study the storm surge caused by 1323Fitow. The timing of Jason-2's passage was approximately 04:50 UTC on 7 October 2013; the Jason-2 altimeter track-240 SLAs of cycles 774 (06:58 UTC on 27 September 2013), 775, and 776 (02:55 UTC on 17 October 2013) are shown in Figure 4b. The SLAs recorded by the Kanmen and Shipu tide gauge stations are depicted in Figure 5. The highest SLA of 1.67 m was recorded by the Kanmen tide gauge station at 13:00 on October 6. The Jason-2 altimeter satellite observed a magnitude of 0.29 m for the storm surge, which was in close agreement with the Kanmen tide gauge reading of 0.29 m (Figure 5). The nearest point on track-240 to the shore was approximately 5 km away from Kanmen, and the similarity between the two observations demonstrates the ability of the altimeter to detect storm surges.

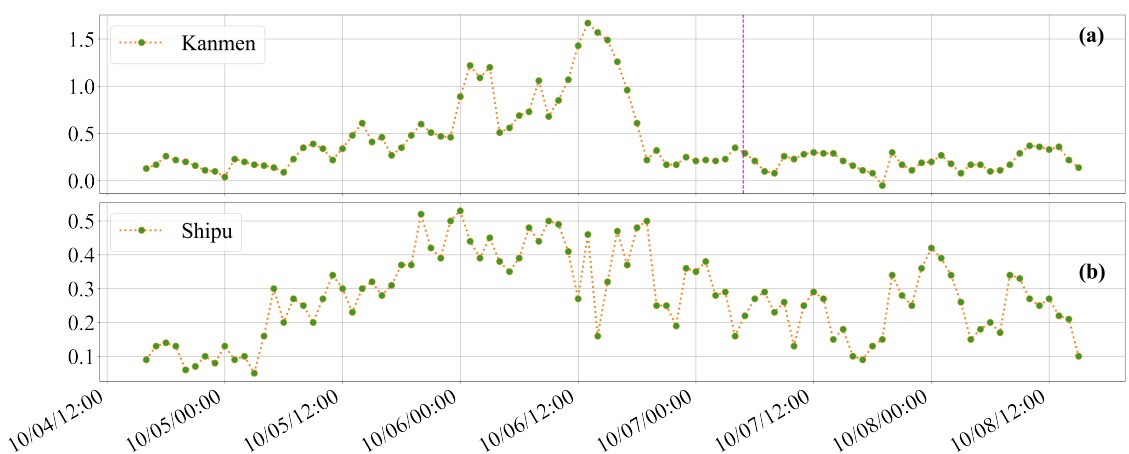

**Figure 5.** De-tided SLAs at (**a**) Kanmen and (**b**) Shipu tide gauge stations. The magenta dotted line indicates the time of the Jason-2 satellite passing (04:50 UTC on 7 October 2013). The landing time of 1323Fitow was 05:15 on 6 October.

### 3.2. Propagation of the Storm Surge Estimated by Jason-2 and Tide Gauges

It can be seen from Figure 4 that the SSHA data exhibit significant oscillations, mainly due to the resolution of 6–7 km. We estimated the cross-shelf decay scale by fitting the altimetric surge height profile (Cycle-774) over selected coastal segments to an exponential function using Equation (2). The segments in Table 1 have a starting latitude of 22.2°N and

different ending latitudes. The cross-shelf decay scale was found to have an average of 102.8 km with a standard deviation of 5 km. The average extrapolated surge magnitude at the coastline was estimated to be 0.63 m, which is relatively close to the actual surge magnitude recorded by the Xiamen tide gauge station (0.66 m).

**Table 1.** Cross-shelf decaying scale and coastal surge by fitting Jason-2 data for Typhoon 1319Usagi.

| Typhoon | Latitude (°N) | L (km) | Coastal Surge (m) |
|---|---|---|---|
| 1319Usagi | 23.71 | 106.53 | 0.62 |
| | 23.66 | 110.93 | 0.62 |
| | 23.61 | 107.87 | 0.62 |
| | 23.55 | 99.17 | 0.64 |
| | Mean | 102.8 | 0.63 |

In addition, the speed of the storm surge propagation was estimated to be approximately 6.1 m/s, calculated using Equation (3) with the Coriolis parameter $f$ of $5.9302 \times 10^{-5}$ s$^{-1}$ at 24°N. The speed was further confirmed by determining the lagged correlation coefficients among the three tide gauge stations (Figure 6a). The highest lagged correlation coefficient was observed between Xiamen and Zhelang with a maximum of 13 h, which was also seen between Longhai and Zhelang. Furthermore, the average propagation speeds of these two sections were calculated using the time lags and along-coast distances (324 km and 319.6 km), yielding estimated speeds of approximately 6.9 m/s and 6.8 m/s, respectively. The cross-shelf e-folding scale was also estimated to be approximately 110 km. These estimates from the altimetry data are consistent with those calculated from the tide gauge data.

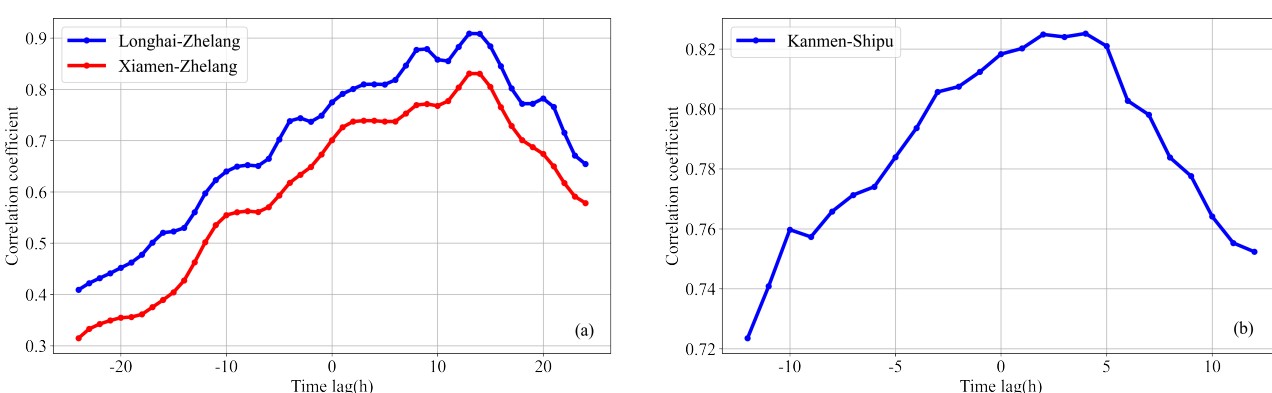

**Figure 6.** The lagged correlation coefficients between the selected tide gauge stations for (**a**) Xiamen and Zhelang, Longhai and Zhelang, and (**b**) Kanmen and Shipu. The lagged correlation coefficients have a maximum value of greater than 0.8.

The water depth and topography in the South China Sea are shown in Figure 2a, with the solid line representing the ETOPO1 water depth and the dashed line representing the ideal water depth along the ground track of the Jason-2 track-88. Figure 2b shows the South China Sea section with $h_1 = 160$ m, $h_2 = 2800$ m, $L_1 = 1.75 \times 10^5$ m, and $L_2 = 1.75 \times 10^5$ m. According to the South China Sea model, the propagation speed of the continental shelf wave was estimated to be around 8.1 m/s. The propagation speed of the coastally trapped waves was estimated to be 6–7 m/s based on both tide gauges and altimeter data, which was significantly slower than the speed ($\sim$22 m/s) of the coastally trapped Kelvin wave ($\sqrt{gh}$, with g = 9.8 m/s) in this region. This difference was attributed to the shallower average water depth (50 m) in the region as stated in [21]. Table 2 summarizes the estimates of the propagation speed for Typhoons 1319Usagi and 1323Fitow using different methods. From this, the examination of the altimetric data suggested that the storm surge propagation was likely a first-mode continental shelf wave.

**Table 2.** Estimates of storm surge propagation speeds for Typhoons 1319Usagi and 1323Fitow using different methods.

| Typhoon Name | Methods | | | |
| --- | --- | --- | --- | --- |
| | Gauge-Derived Tide | Altimetry-Derived | Continental Shelf Wave | Kelvin Wave ($\sqrt{gh}$) |
| 1319Usagi | ~6.8 m/s | 6.1 m/s | 8.1 m/s | 22 m/s |
| 1323Fitow | ~10 m/s | 8.0 m/s | 9.3 m/s | 22 m/s |

For Typhoon Fitow in 2013, the speed of the coastally trapped waves was also examined based on observations from tide gauges and Jason-2 altimetric data. Firstly, the lag between the surge peaks in Kanmen and Shipu was 4 h (Figure 6). The along-coast distance was 142 km. Secondly, it was determined that the coastally trapped wave speed was estimated to be ~10 m/s. The cross-shelf e-folding scale was calculated to be ~146 km (the Coriolis parameter of $6.8449 \times 10^{-5}$ s$^{-1}$ at 28°N). Thirdly, the altimetric surge height profile (cycle-775) in the form of a dome-like shape over the inner shelf, observed after the typhoon landfall, should not be used for estimating the length scale or propagation speed due to the presence of the offshore wind-driven Ekman transport causing a drop in the coastal sea level from the earlier peak surge. We excluded nearshore data and attempted to fit the altimetric data that conformed to an exponential distribution (coastal segments within the latitudinal range of 25.9–27.6°N); the average propagation speed of the surge was determined to be 8.0 m/s and the cross-shelf e-folding scale was estimated to be 116.66 km on average (Table 3). Finally, according to the East China Sea model, the parameters $h_1 = 190$ m, $h_2 = 2200$ m, $L_1 = 3.8 \times 10^5$ m, and $L_2 = 1.2 \times 10^5$ m resulted in a continental shelf wave propagation speed of about 9.3 m/s. The comparisons indicated that the storm surge characteristics matched those estimated by the tide gauge data. Therefore, this storm surge likely propagated in the form of a first-mode continental shelf wave.

**Table 3.** Cross-shelf decaying scale and coastal surge; fitting Jason-2 data for Typhoon 1323Fitow.

| Typhoon | Latitude (°N) | L (km) | Coastal Surge (m) |
| --- | --- | --- | --- |
| | 27.76 | 141.25 | 0.77 |
| | 27.71 | 114.16 | 0.89 |
| 1323Fitow | 27.66 | 94.57 | 1.05 |
| | Mean | 116.66 | 0.90 |

### 3.3. Accuracy Evaluation of Remote Sensing SST Fusion Data

The QF206 buoy was deployed close to the shore in the Southern China Sea, and data were collected at 30 min intervals. After the storm, a 4.6 °C drop in the SST was observed (Figure 7). It was reported in previous studies that before a typhoon arrives, the SST rises and the wind speed decreases, known as the "typhoon cooling effect" [33]. This occurs due to the cooling impact of the strong winds and heavy rain of the storm on the sea surface and the surrounding area's wind weakening. The SST did not return to its original level after the storm, as the net heat flux around the surface of this area often weakened in late September (Figure 7).

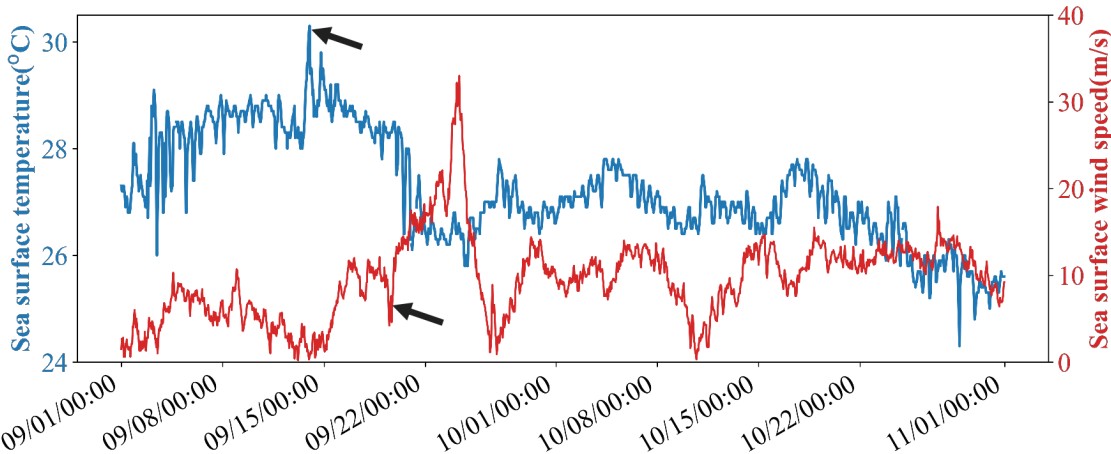

**Figure 7.** Temporal sea surface temperature and wind speed from no. QF206. The temperature dropped from 30.4 °C to 25.8 °C between 09:00 13 September and 06:30 22 September. The "typhoon cooling effect" is indicated by the black arrow.

The in situ data from the buoy provides valuable information for studying the thermal and dynamic changes caused by Typhoon 1319Usagi and verifying the G1SST fusion data. As shown in Figure 8, there is a comparison between the daily SSTs from G1SST (represented by the blue line) and the buoy measurements (represented by the red line). The results show a root mean square difference of 0.4 °C and a correlation coefficient of 0.93 between 1 September and 31 October, indicating a good agreement between the two data sources. Both the G1SST and the buoy data captured the cooling of the SST.

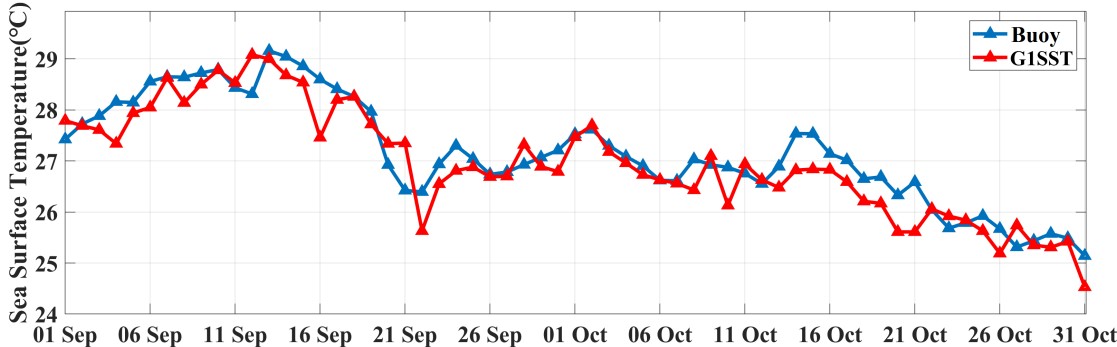

**Figure 8.** Comparison of the sea surface temperature between the observation (blue) and G1SST (red) from 1 September to 31 October.

### 3.4. SST Response to the Typhoon

According to the remote sensing SST fusion data obtained from G1SST, Typhoon 1319Usagi caused a drop in the daily average SST from 29 °C to 25.63 °C, as shown in Figure 8. Meanwhile, the buoy data showed a decrease from 29.16 °C to 26.4 °C from 13 September to 22 September, as seen in the same figure. These findings indicate that the fusion data from G1SST are accurate and reliable for analyzing the oceanic response to typhoons.

To analyze the spatial distribution of SST changes induced by Typhoon 1319Usagi, we first presented the wind speeds during 13–24 September (obtained from ERA5). As shown in Figure 9, there was no significant change in wind speed during 13–15 September. Daily SSTAs were computed by subtracting climatological daily SST values. To more accurately describe the SSTA conditions during this period, we calculated the average SSTA during 9–15 September as the pre-typhoon SSTA state. The final SSTAs were calculated

by subtracting the pre-typhoon SSTA states. This methodology allowed us to capture the temporal evolution of SST changes before, during, and after the typhoon passage.

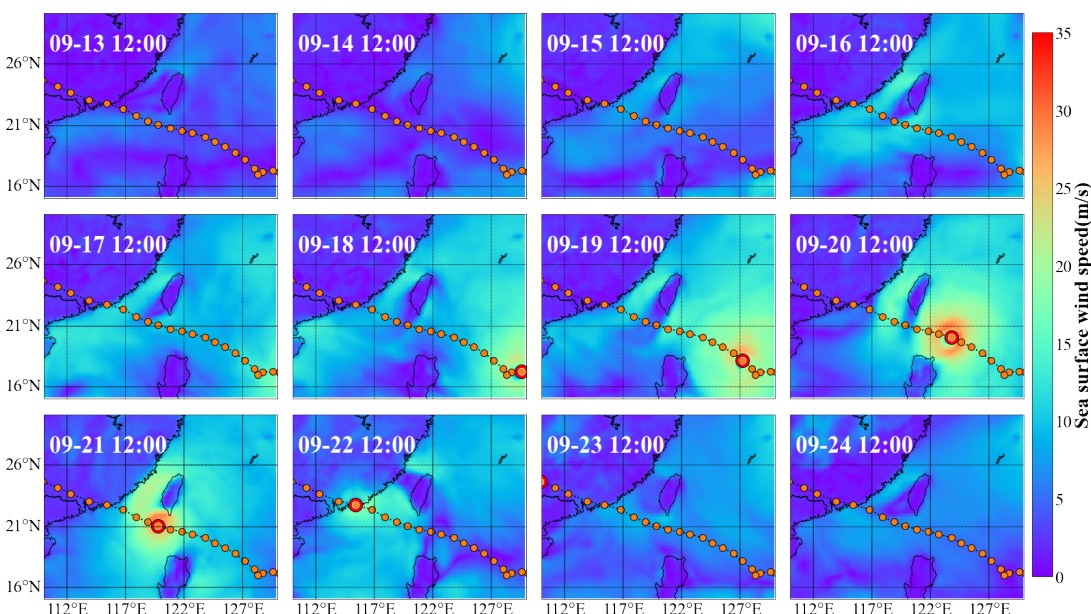

**Figure 9.** Time evolution of the daily sea surface wind speed from 13 September to 24 September. The big circle in red is the location of the typhoon at 12:00 that day.

The SSTAs were calculated from 19 September to 24 September compared to the average SSTA from September 9 to 15. The cooling shown in Figure 10 may not always align closely with the typhoon trajectory due to the potential influence of various factors, including ocean currents and atmospheric conditions. For instance, the observed cooling on September 20 could have been impacted by local oceanic and atmospheric conditions, such as changes in the intensity of the typhoon in each quadrant or wind direction. On 21 September, the SST decreased by 4.5 °C compared to the mean SSTA from September 9 to 15. This decrease was more pronounced on the right side of the typhoon track (blue color), which aligns with previous research findings [2,30,34,48].

### SST change(°C) relative to mean(09/09/2013-09/15/2013)

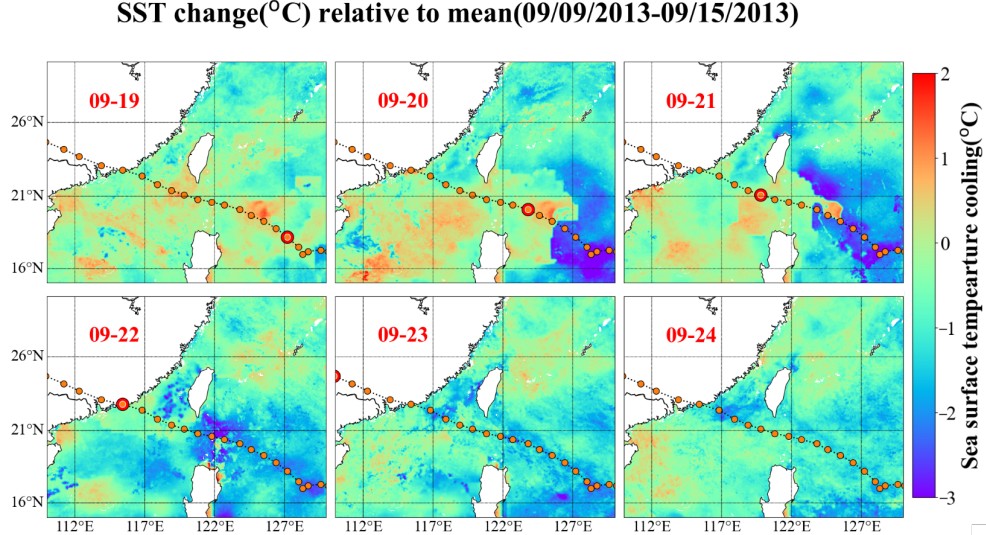

**Figure 10.** The daily sea surface temperature change from 19 September to 24 September relative to the mean sea surface temperature anomalies from 9–15 September during Typhoon 1319Usagi. The big circle in red is the location of the typhoon at 12:00 that day.

In the context of Typhoon 1323Fitow, ERA5 reanalysis wind speeds were also used to determine the time period of the pre-typhoon state. Figure 11 indicates that the wind speed in the entire region did not display significant variations before 1 October. Accordingly, we selected the period between 25 September and 1 October as the pre-typhoon state, calculated the average SSTA, and leveraged this value to calculate the final SSTA. This approach was adopted to better understand the changes and trends in the ocean temperature during the typhoon period.

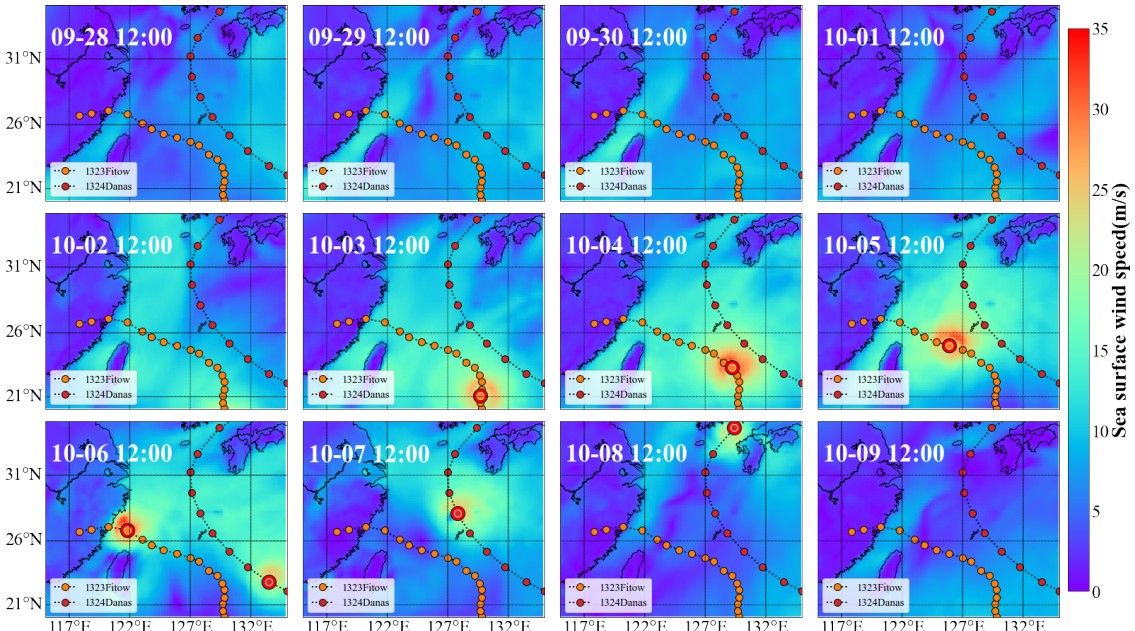

**Figure 11.** Time evolution of the daily sea surface wind speed from 28 September to 9 October during Typhoon 1323Fitow. The big circle in red is the location of the typhoon at 12:00 that day.

An intriguing observation can be made from Figure 12, i.e., the cooling effect caused by Typhoon 1323Fitow was larger than that caused by Typhoon 1319Usagi. This can be attributed to the historical data of typhoons, which demonstrated that 1323Fitow and 1324Danas simultaneously impacted the Southeast China coast from 6 to 8 October, leading to a substantial temperature drop. In light of the G1SST data, the response of the surface temperature to Typhoon 1323Fitow was investigated. The daily SSTs in the region of 116∼130°E and 22∼33°N were analyzed, revealing a decrease in the SST of approximately 1.7 °C, from 27.7 °C to 26.0 °C. Furthermore, analogous to Typhoon 1319Usagi, the SST did not recover to its original level after the storm. The most significant cooling effect was observed on 5 October, the SST dropped by 4.0 °C, and the cooling region was situated on the right side of the typhoon's trajectory. The SSTA distribution on 6 October demonstrated an apparent temperature drop zone along the right side of the typhoon's track.

In conclusion, the impact of Typhoon 1323Fitow on SST was substantial, resulting in a larger cooling area compared to Typhoon 1319Usagi. This underscores the importance of continued monitoring and the analysis of a satellite-based SST response to typhoons. The results of this study have implications for better understanding the relationship between typhoons and SST, which could have significant implications for the protection of marine ecosystems and the livelihoods of coastal communities.

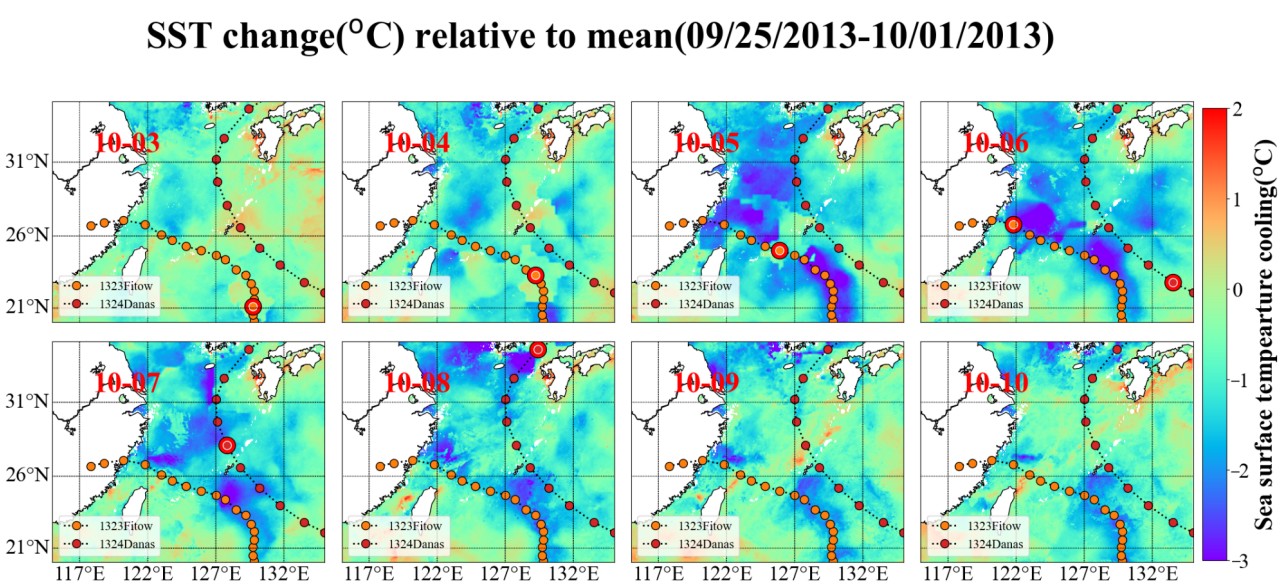

**Figure 12.** The change in daily sea surface temperatures from October 3 to 10 relative to the pre-typhoon state. The big circle in red is the location of the typhoon at 12:00 that day.

## 4. Discussion

Our study provides a more comprehensive understanding of the oceanic responses to typhoons in the Southeast China Sea by using a combination of different remote sensing tools and in situ observations. Moreover, our research emphasizes the importance of studying the oceanic responses to typhoons off the coast of the Southeast China Sea. While previous studies have focused on other regions, such as the Gulf of Mexico, our study contributes to a better understanding of how typhoons affect oceanic responses in this unique region. Our findings show that typhoons can create storm surges and propagate in the Southeast China Sea, similar to other regions; we analyzed sea surface cooling using remote sensing measurements, which is an important consideration for future disaster risk management in the region.

There are still several areas that require further investigation. One of the major limitations is the sparse spatial distribution of tide gauges and buoys in the region, which could result in some uncertainties in our analysis. Additionally, eddies can affect the storm surge by modulating the vertical structure of the oceanic boundary layer, which in turn influences the storm surge and surface cooling. Therefore, it is important to consider that the influence of mesoscale eddies in the analysis could provide a more comprehensive understanding of the oceanic response to typhoons. Finally, future studies could examine the impact of typhoons on storm surges and SST by numerical simulation.

## 5. Conclusions

Our study provides new insights into the oceanic responses to typhoons off the coast of the Southeast China Sea. By combining data from multiple sources, including altimeters, remotely sensed fusion data, tide gauges, and buoys, we were able to accurately monitor and analyze the peak storm surge, cross-shelf e-folding decay scale, and propagation speed of Typhoons 1319Usagi and 1323Fitow. Our analysis revealed that 1319Usagi and 1323Fitow created continental shelf waves close to the Taiwan Strait.

In addition, our study highlights the importance of utilizing remote sensing data in understanding the oceanic responses to typhoons. The high correlation coefficient and low RMSE between the G1SST data from GHRSST and the buoy measurements showed the effectiveness of combining multiple sources of data. Moreover, our study revealed that the maximum SST cooling occurred on the right side of the typhoon's track and that the surface temperature did not recover to its original level after the storm. Our findings demonstrate the need for continued research into the oceanic responses to typhoons in the Southeast

China Sea, and the importance of utilizing multi-source remote sensing data for a more comprehensive understanding of these events.

**Author Contributions:** G.H. and J.Y. conceived the idea and designed the approach. X.L. analyzed the data. X.L. and C.W. wrote this paper. G.H., C.W. and J.Y. contributed to the writing, editing, and proofreading. All authors have read and agreed to the published version of the manuscript.

**Funding:** This work was supported by the Scientific Research Fund of the Second Institute of Oceanography, Ministry of Natural Resources of China (grant no. JB2205), funded by the China Postdoctoral Science Foundation (grant no. 2022M723705), preferential support for postdoctoral research projects in Zhejiang Province (grant no. ZJ2022041), and the Innovation Group Project of Southern Marine Science and Engineering Guangdong Laboratory (Zhuhai) (no. 311021004), supported in part by the National Natural Science Foundation of China (grants 42176024 and 42090040).

**Data Availability Statement:** Not applicable.

**Conflicts of Interest:** The authors declare no conflict of interest.

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
