# Peer review of "Remote Sensing Analysis of Typhoon-Induced Storm Surges and Sea Surface Cooling in Chinese Coastal Waters"

_remotesensing, doi:10.3390/rs15071844_

Round 1

Reviewer 1 Report

The application of altimeter data to the monitoring of typhoon-induced storm surge near shore is an interesting topic. The authors have also verified the accuracy of the altimeter data using tide gauges data, making the conclusions very convincing. I believe it is worthy of publication in RS, but there are several queries that need to be answered or corrected by the authors before it can be formally published.

Lines 51-56: The purpose of the study presented here is slightly far-fetched, please rephrase it, e.g. the success of the altimeter in observing typhoon storm surges is highly stochastic, yet ......

Lines 68-72: I don't think this kind of high resolution SST fusing multiple remote sensing observations is necessarily better than a single coarse resolution remote sensing SST, and the fusion process itself introduces errors.

Line 113: our previous studies indicated ......

Lines 252-306: The propagation speed estimated by different means are tabulated so that the reader can quickly grasp the information

The cooling shown in Figure 12 is sometimes not close to the right side of the trajectory, such as September 20-21, please explain.

Figure 14-15, it is confusing, obviously the area you are talking about is beyond the area shown in Figure 13

Reviewer 2 Report

Manuscript Number: remotesensing-2234793

Title: Remote Sensing Analysis of Tropical Cyclone-induced Storm Surge and Sea Surface Cooling in Chinese Coastal Waters

Authors: Xiaohui Li, Guoqi Han, Jingsong Yang, Caixia Wang

The authors have analyzed the remotely sensed altimeter and satellite measurements to examine the impacts of four typhoons on storm surge and sea surface cooling within the Chinese coastal waters.

At the present stage the article needs Major revisions. Recommended for Major Revision.

Comments:

1.          “Typhoons” should be used throughout the paper.

2.          Figure 1 needs to be revised. What does this ‘blue star’ indicate ? Please label Yellow Sea, East China Sea and South China Sea (as these terms have been used in the text). The caption can also be revised accordingly to bring more clarity on the description of the cyclone tracks.

3.          The authors have used equation 1 to specifically identify the correction terms. Can the authors provide a reference to this equation.

4.          The authors have calculated SST drop after the cyclone. There may be some issues with the calculation of the SST drop. If the authors can bring more clarity on how they have calculated the SST drop.

5.          If the authors can explain the presence of mesoscale eddies, coastal Kelvin waves within the SSHA datasets and translational speed. How these can dynamically impact the storm surge and SST variations along the cyclone track ? 

Reviewer 3 Report

-       As the authors mention in the introduction, the use of altimetry and SST remote sensing data for TC research is not new. They state that the goal of the paper is to apply these to a series of TCs that hit China’s coast, but other than being a case study, it is unclear what new results are gleamed from this. If it is a case study of these specific TCs, then more information needs to be provided about previous studies of these storms. If it is a general discussion of TC storm surge impacts on this coast, then more detail needs to be provided on that topic. In general, the wording needs to make it abundantly clear to the reader what is novel about this research and the introduction should be expanded to include these topics rather than merely mentioning previous studies that have used remote sensing to analyze TCs.

-       Figure 4, 6, 7: It would be useful to also mark when the TCs passed these regions on the figure or in the caption, even if the date range on the figure encompasses the TC. This will help the reader better understand and interpret the figure.

-       Figure 5: It would be helpful for the typhoon name to be included on the figure, not just in the caption

-       Section 3.1.2: the Butterworth low-pass filter should be described in methods, not in your results

-       The paper is oddly organized and jumps around, analyzing figures in later sections that were presented early on. This can be confusing and frustrating for a reader. Paper should be better organized for flow and easier comprehension of results.

-       Figure 10 – tics on x-axis make it nearly impossible to tell which day corresponds to which tic

-       General: Figures should specify which typhoon(s) they correspond to

-       The conclusions feel very brief and underdeveloped. Yes, kelvin waves, continental shelf waves, storm surge, etc. are important to monitor and can be monitored with a combination of satellites and in situ observations, as has been demonstrated in previous studies. It remains unclear from this paper what new information is truly gleamed from this paper. If it is intended as a case study comparing these three storms, then more emphasis needs to be placed upon this and corresponding background work and analysis should be conducted. As it stands, this feels incomplete and lacks impact.  

Round 2

Reviewer 2 Report

The authors have responded well to my comments. However, I still have a doubt about the calculation of the SST. The article can be accepted for publication subjected to the following revisions.

1. I still have a concern about whether the authors have removed the seasonal cycle from the SST data. This needs to clarify as the authors are looking into the SST drop and storm surge due to the cyclone.

The authors should follow the reference below to calculate the SST drop. 

Mandal et al. (2018). https://doi.org/10.1007/s00024-018-1932-8 
